# A Review for Artificial Intelligence Based Protein Subcellular Localization

**DOI:** 10.3390/biom14040409

**Published:** 2024-03-27

**Authors:** Hanyu Xiao, Yijin Zou, Jieqiong Wang, Shibiao Wan

**Affiliations:** 1Department of Genetics, Cell Biology and Anatomy, College of Medicine, University of Nebraska Medical Center, Omaha, NE 68198, USA; haxiao@unmc.edu; 2College of Veterinary Medicine, China Agricultural University, Beijing 100193, China; 2020305010318@cau.edu.cn; 3Department of Neurological Sciences, College of Medicine, University of Nebraska Medical Center, Omaha, NE 68198, USA; jiwang@unmc.edu

**Keywords:** protein subcellular localization, machine learning, deep learning, artificial intelligence, gene ontology, sequence analysis

## Abstract

Proteins need to be located in appropriate spatiotemporal contexts to carry out their diverse biological functions. Mislocalized proteins may lead to a broad range of diseases, such as cancer and Alzheimer’s disease. Knowing where a target protein resides within a cell will give insights into tailored drug design for a disease. As the gold validation standard, the conventional wet lab uses fluorescent microscopy imaging, immunoelectron microscopy, and fluorescent biomarker tags for protein subcellular location identification. However, the booming era of proteomics and high-throughput sequencing generates tons of newly discovered proteins, making protein subcellular localization by wet-lab experiments a mission impossible. To tackle this concern, in the past decades, artificial intelligence (AI) and machine learning (ML), especially deep learning methods, have made significant progress in this research area. In this article, we review the latest advances in AI-based method development in three typical types of approaches, including sequence-based, knowledge-based, and image-based methods. We also elaborately discuss existing challenges and future directions in AI-based method development in this research field.

## 1. Introduction

Within a cell, mature proteins must reside in specific subcellular structures to properly perform their biological roles, as different cellular compartments provide distinct chemical environments (e.g., pH and redox conditions), potential interacting partners, or substrates for diverse functions [1,2]. Most cellular biological processes, such as the nucleocytosolic shuttling of transcription factors [3], the relocalization of mitochondrial proteins during apoptosis [4], and the endocytic uptake of cell-surface cargo receptors, all rely on precise protein localization. Conversely, mislocalization is often associated with cellular dysfunction and diseases, such as cancer [5,6], neurodegenerative diseases [7,8], and metabolic disorders [9,10]. 

Conventionally, identifying subcellular localization of proteins primarily relies on wet lab experimental methods. Fluorescence microscopy imaging, which applies fluorescent dyes or fluorescent protein tags to label target proteins, has commonly been used for observing their distribution within cells [11,12]. This method has become one of the preferred tools for studying protein subcellular localization due to its high resolution and real-time observation advantages [13]. By using labeled antibodies against target proteins, the immunoelectron microscopy technique is regarded as a gold standard to provide the high resolution of electron microscopy [14]. Another method involves the use of fluorescent biomarker tags [15] like the protein A-GFP tag, which fuses a fluorescent protein with the target protein, allowing it to emit a fluorescent signal among different cell compartments [16]. These experimental methods yield high-resolution location of targeted proteins for researchers, enabling direct observation to uncover biological processes and metabolic mechanisms.

However, these wet lab experimental methods also have some significant drawbacks: they often require expensive equipment and time-consuming steps, making them costly for large-scale studies. These problems are exacerbated given that the number of newly discovered proteins has increased exponentially in the post-genomic era. Take the UniProt Database [17] as an example. The gap between the reviewed and unreviewed proteins has significantly expanded during the past decade (Figure 1A). Specifically, as shown in Figure 1B, in the latest 2024_01. version of UniProt, a notable majority of data entries are unreviewed proteins in TrEMBL. In this case, implementing wet lab experiments alone for subcellular localization determination for remarkably large amounts of data from different species (Figure 1C) becomes an impossible mission. Moreover, the rich collection of accurately annotated protein data in databases (Figure 1D) can facilitate the development of robust prediction methods. It is noteworthy that, compared to TrEMBL, the smaller size of Swiss-Prot can be attributed to the rigorous manual curation of proteins. Conversely, TrEMBL comprises computationally analyzed records, leading to a plethora of protein sequences awaiting annotation before being entered into Swiss-Prot. Fortunately, the necessity of manual curation might be alleviated if transcript-translated sequences can be validated through proteomics detection. An example of such an approach can be observed in the Human Protein Atlas (HPA) [18,19], as we will elaborate in subsequent sections, where RNA-seq data were employed to corroborate immunofluorescence subcellular localization findings. In this context, leveraging computational models, particularly AI-assisted methodologies renowned for their adeptness in handling large-scale datasets, can offer substantial benefits.

Recent decades have witnessed the booming of in silico methods for protein subcellular location prediction. Based on features used for computational modeling, most existing methods can be generally divided into three main categories: (1) sequence-based methods, which only use the amino acid sequence of the query protein as inputs; (2) knowledge-based methods, using protein annotations from multiple databases to correlate the information with their subcellular locations; and (3) image-based methods, extracting subcellular location features from bioimages and then identifying the likelihood of proteins being located in various subcellular compartments. The primary sequence for a protein is much easier to obtain with existing sequencing technologies. With remarkable advances in machine learning and deep learning, coupled with an increasing number of proteins with experimentally determined localization information as well as functional annotations and imaging records in publicly available databases, accurate and efficient computational frameworks provide a promising way for protein subcellular localization. 

In this review, we will first present some remarkable progress in in silico models, including the three major types of models mentioned above. In Section 2, we will introduce common features and algorithms used in sequence-based methods and also for knowledge-based and image-based frameworks in Section 3 and Section 4, respectively. The simplified flowchart for the prediction frameworks mentioned is illustrated in Figure 2. In Section 5, we will give an overview of protein subcellular localization models that are specially designed for different species. Lastly, we will explore the existing challenges and future trajectories of this research domain and propose our expectations.

## 2. Sequence-Based Methods

Sequence-based methods directly use the amino acid sequence of a query protein as model inputs and attempt to find the correlations between protein sequences and their subcellular locations. With the advent of high-throughput sequencing technologies, large-scale genomic and proteomic data are easily obtained, allowing new big-data-based models to be constructed. In addition, as proteins consist of sequences of amino acids, they are fit for computational models that extract features for subcellular localization. However, protein sequences might not capture full information for protein subcellular localization, particularly in the cases of protein post-translational modifications or protein dynamics processes within cells once the protein is synthesized, which may influence where proteins reside.

### 2.1. Sequence-Based Features

In protein primary sequences, the 20 standard amino acids (AA) exert different biochemical properties such as hydrophobicity, hydrophilicity, side-chain characters, etc. Sequence-based methods intend to make predictions out of the correlations between protein subcellular locations and the information embedded in amino acid sequences. There are three major types of features used for model construction: AA composition information, sorting signal information, and evolutionary information.

The composition-based features, which include AA occurrences and order in the query sequence, were commonly used in the earliest subcellular prediction methods. Moreover, previous studies have confirmed a better performance of the model by combining AA original sequence, gapped amino acid composition (GapAA) [20], and amino-acid-pair composition (PairAA) [21]. Based on AA-composition features, Chou [22] proposed pseudo-amino-acid composition (PseAA) using the sequence-order correlation factor for greater biomedical property discovery when avoiding the high-dimensional vector formation. The simplicity of composition features helps the generalization and interpretation of the computational models since they capture the most basic trends in protein sequences associated with their locations. However, they may not provide sufficient resolution for a high accuracy rate, since there is a loss of information about important sequences or structural motifs highly related to proteins’ subcellular location.

The sorting signal sequences or signal peptides, including transit peptides like mitochondrial transit peptides (mTPs) and chloroplast transit peptides (cTPs) [23], are short and cleavable segments of amino acid sequences added to newly synthesized proteins, determining their destination in the transportation process. These short peptides possess the directions mature proteins should be transported, reflecting the possible location event for one protein [24]. Available approaches with signal peptides for protein localization mainly refer to finding their cleavage sites [25]. As described in previous studies, sorting-signal sequences vary in length and composition but have similar structures: the N-terminal flanking region, also known as the n-region, the central hydrophobic region (h-region), and the C-terminal flanking region (c-region) [26]. The hydrophobicity in the h-region and a large proportion of nonpolar residues in the c-region are used to label the cleavage sites by computational methods [27,28]. According to the location signal embedded in those short peptides, one can mimic the de facto information processing in cells and find the target spot of the test protein. 

In addition, based on the fact that homologous sequences are likely to share the same subcellular location, the unknown protein can be assigned the same subcellular location as its homologs generated from PSI-BLAST [29]. Moreover, the evolutionary similarity profiles extracted from the position-specific scoring matrix (PSSM) and position-specific frequency matrix (PSFM) derived from multiple sequence alignment results can contribute as classification features providing valuable information such as conserved motifs or targeting signals among different protein families. This representation can also be extended by integrating pseudo-analysis [30]. Once aligned with known homologs in the database, this method can achieve high accuracy. However, as one amino acid change can directly influence the characters of one protein sequence, this method is more likely to be one of the sources of the feature basis of prediction models.

### 2.2. Sequences-Based AI Approaches

Most computational frameworks include three major steps: feature extraction, feature selection, and final classification. Considering common features discussed above, the complexity of the models developed also increases with the amount of data processed and the dimension of input features, from traditional machine learning classification to complex deep learning analytical models. Besides the development of computational frameworks, we will also introduce techniques that are used to improve the algorithms dealing with multi-location proteins in the following. 

For conventional classification, the Support Vector Machine (SVM) [31], K-Nearest Neighbor (KNN) [32], and Random Forest (RF) [33,34] are widely chosen classifiers for training. Their simplicity makes them easy to use for prediction protocols with fast speed and low computational cost, suitable for limited data and low-dimensional inputs. Combined with highly efficient feature extraction methods, these frameworks will work well in most cases [35]. For instance, Du et al. [36] proposed two novel feature extraction methods that utilize evolutionary information via the transition matrix of the consensus sequence (CTM) and PSSM before adopting SVM, which, in the end, reached an overall accuracy of 99.7% in CL317 dataset. A feature-extraction-based hierarchical extreme learning machine (H-ELM) introduced by Zhang et al. [37] can handle high-dimension feature inputs directly without demanding dimension reduction for acceptable results. Alaa et al. [38] exploits an extended Markov chain to provide the latent feature vector, which records micro-similarities between the given sequence and their counterparts in reference models. These methods help extract more abundant features of query sequences and provide better performance. 

However, these conventional models may not perform well in complex scenarios [1], especially multi-locational protein prediction [30]. Though many proteins only stay in one subcellular space, studies have discovered many multi-location proteins that have special functions or are involved in crucial biological steps [39]. Moreover, rather than staying in one place, proteins move from one subcellular compartment to another or simultaneously reside at two locations and participate in different cellular processes [40]. Recent studies have also shown the remarkable significance of multilocation proteins in cell growth and development [41]. For instance, phosphorylation-related multilocation proteins can function as a “needle and thread” via protein–protein interactions (PPI), thus playing an important role in organelle communication and regulating plant growth [42]. Under these circumstances, there are mainly two ways for predicting multi-location proteins based on conventional classifiers: algorithm adaption and problem transformation. The former method extends existing algorithms to deal with multi-label problems. Jiang et al. [43] considers weighted prior probabilities with a multi-label KNN algorithm to increase the model accuracy. Library of SVM (LIBSVM) toolbox [36,44], instead, uses a one-versus-one (OVO) strategy to solve multi-class classification problems. Customization of well-known algorithms enhances their ability for specific requirements but there is a risk of overfitting and it may require significant computational resources. The problem transformation approach focuses on transforming the original problem into a different representation or formulation that is solvable with existing algorithms [45,46], such as converting a multi-location classification problem into multiple single-label classification problems [47]. Shen et al. [30] introduces multi-kernel SVM by training multiple independent SVM classifiers to solve single-label problems before combining their results, one classifier for each class. Following this idea, an algorithm can be easily extended to solve multi-label classification.

In summary, traditional machine learning algorithms can achieve fast training times and high accuracy in scenarios with well-organized feature spaces and clear decision boundaries; their performance may degrade quickly when faced with large-scale data inputs, even with tailored classifiers featuring more selected features. Dimension reduction [48] and parallel processing [49] can be applied to mitigate the challenges, allowing an improved computational method scalability.

As multi-layered structure provides better performance compared to traditional approaches [33], more methods based on deep networks, especially neural networks, have become increasingly popular in protein subcellular localization research [50,51]. Starting as effective feature extractors which automatically obtain deep features embedded in sequences [52], convolutional neural network (CNN) is widely implanted in multi-locus protein localization framework. Mining deeper, Kaleel et al. [53] ensemble Deep N-to-1 Convolutional Neural Networks that predict the location of the endomembrane system and secretory pathway versus all others and outperform many state-of-the-art web servers. Cong et al. [54] proposed a self-evolving deep convolutional neural network (DCNN) protocol to solve the difficulties in feature correlation between sites and avoid the impact of unknown data distribution while using the self-attention mechanism [55] and a customized loss function to ensure the model performance. In addition, a long short-term memory network (LSTM) which combines the previous states and current inputs is also commonly used [56,57], with Generative Adversarial Network (GAN) [58] and Synthetic Minority Over-sampling Technique (SMOTE) [59] used for synthesizing minority samples to deal with data imbalance. Developing data augmentation methods by deep learning algorithms has also made protein language model construction possible [60,61]. Through transfer learning [62], pretrained models can be fine-tuned on different downstream tasks, reducing the need for large amounts of labeled data for training. For example, Heinzinger et al. [63] proposed Sequence-to-Vector (SeqVec) that embeds biophysical properties of protein sequences as continuous vectors by using the natural language processing model ELMo on unlabeled big data. This represents a way to speed up the prediction process independent of the size of inputs. As protein sequences can also be tokenized and coded with a certain pattern as natural languages [64], some well-developed models (e.g., Univeral Language Model Fine-tuning (ULMFiT)) [65] have also been repurposed to protein-related questions, like AlphaFold [66] for protein 3D structure prediction and ProteinBERT [67] for bidirectional sequence modeling and Gene Ontology annotation prediction. ESM2 [67], the pretrained deep language model inspired by Bidirectional Encoder Representations from Transformers (BERT) [68], allows fast training of localization predictors with limited labeled protein sequence data [69]. Transformer-based architectures leverage self-attention mechanisms, enabling the capture of key hidden vectors for subcellular localization from sequences. This intrinsic ability facilitates information exchange across all positions without necessitating pooling operations like CNN, greatly enhancing the ability of extracting deeper information. However, this will take more time and larger computational resources for training to gain higher model performance, since the results may be similar to simple classifiers when the prediction scale is small [60]. Details of the computational models mentioned above can be found in Table 1. 

Deep learning will demonstrate exceptional outcomes dealing with high-dimensional inputs with deep feature extraction, eliminating the need for manual feature engineering and capturing intricate patterns in sequences. However, large, labeled, and high-quality datasets are still needed for original model training, which results in too many hyper-parameters and makes it hard to interpret the model itself [33].

## 3. Knowledge-Based Methods

There is a strong correspondence between annotations and subcellular locations of proteins. Knowledge-based methods for protein localization prediction mainly extract information from annotation databases and convert them into numeric features as model inputs. Since the annotations are generated based on biological processes, functions, or protein interactions within cells, models can provide more interpretable results for subcellular localization. But knowledge data are limited and only applicable to well-curated proteins, which limits the predictive power of this kind of method for novel or newly discovered proteins. In recent studies [75,76,77], different kinds of information are fused together for better model performance, given that computational methods excel with high dimensional data as inputs.

### 3.1. Legitimacy of Using Gene Ontology (GO) Features

Knowledge-based methods tend to dig into the correlation between the annotation of one protein and its subcellular location to establish predictors. Compared to Swiss-Prot keywords [78,79] or PubMed abstracts [80,81], Gene Ontology (GO)-terms-based methods are more attractive for the following reasons.

GO terms describe reviewed knowledge of the biological domain in three aspects: (1) Molecular Function, representing activities that can be performed by individual or by assembled complexes of gene products at the molecular level; (2) Cellular Component, labeling locations relative to cellular compartments; and (3) Biological Process, describing the events achieved by one or more ordered assemblies of molecular functions. This well-organized information can be used for protein subcellular localization because, (1) instead of table-lookup, which is dependent on cellular component GO terms, they perform deeper mining into items to accumulate every related GO category to improve prediction results; (2) the methods outperform previous sequence-based methods without compromising either inputs or outputs [82]. Mining deeper, the GO term itself is structurally organized but loosely hierarchical, consisting of cellular components, biological processes, and molecular functions of gene products. The relationship between GO terms can be “part-of” (part and whole), which may embed some similarity information, and “is-a” (parent and child), which may result in more than one parent term. Starting from semantic similarity measurement, SS-Loc [83] incorporates a richer source of homologs and generates more features for prediction. Making use of the loosely hierarchical structure, relevance similarity (RS) considers the “distance” between the parent and child nodes. Take HybridGO-Loc [2], for example; it combines the frequency of occurrences of GO terms and semantic similarity between extracted GO terms to form a hybridized vector as input features, giving outstanding performance. 

Mapping AA entries of a query protein or accession number (AC) of its homologs to the GO database [84] will result in a list of GO items representing the possible functions and biological metabolism process this protein is involved in. For further computational method implementation, reorganizing and transferring the list of data into numerical vectors is of high significance. Gneg-mPLoc [85], Euk-pLoc [86], and Hum-pLoc [87] consider GO terms as the basis of forming a Euclidean space, which only consists of 0 or 1 for co-ordinates. ProLoc-GO [43], on the other hand, represented the hit of annotated GO terms mined from Gene Ontology Annotation (GOA) with an n-dimensional binary feature vector. The constructed GO vectors are used for the following training. 

### 3.2. Knowledge-Based AI Approaches

Originally, most machine learning methods used GO terms as the only input sources in a simple classification model [88,89]. Given the growing richness of comprehensive protein annotation like related metabolism pathways and structural information, the integration of various input sources, including annotations, interaction networks, and pathway enrichment knowledge, contributes to a multi-view foundation for model improvement [75,90,91]. Applying deep learning algorithms enables a more comprehensive understanding of these high-dimensional and complex features and furthers the combination of sequence and knowledge as input sources. According to the number of input sources, the methods can be roughly divided into GO terms only and fusion methods.

For a single input source, mGOASVM [92] introduces a new decision scheme in SVM multi-class classifiers to collect all the positive decisions, enabling both single- and multi-label localization. AD-SVM [93] enhances the binary relevance methods by integrating an adaptive decision scheme, thereby transforming the linear SVMs into piecewise linear SVMs, reducing the over-prediction instances. By using the frequency of the appearance of one protein in different places, Euk-mPLoc 2.0 [94] creates a virtual sample counting the appearance of protein to separate the total sequence input and the number of locations. However, a large number of proteins, especially newly discovered proteins, have not been functionally annotated yet and directly using homologs cannot guarantee the availability of enough GO terms to be found in the GOA Database. Moreover, the GO is not related to the representation of dynamics or pathway dependencies for protein, which will result in the risk of noise and overestimation of the novel proteins [95]. More details of the methods mentioned can be found in Table 2.

To improve the interpretability of the proposed model, Kyoto Encyclopedia of Genes and Genomes (KEGG) pathways is also considered as a functional annotation that can be incorporated in the computational approaches [96]. Since in vivo protein interaction is likely to reside within the same subcellular locations, it is possible to reveal protein subcellular localization with protein–protein interaction (PPI) networks [97,98,99], which is sensitive to mislocalization events [100]. The BioPlex network [101,102], which systematically explores the human interactome developed from affinity purification–mass spectrometry analyses, has also reflected protein function and localization information. As a multi-scale map (MuSIC 1.0) with 69 subcellular systems of human cells generated from BioPlex and Human Protein Atlas (HPA) data integration by Qin et al. [103], protein interactions can be observed from a spatial dimension, providing rich features for knowledge-based model development.

The fusion methods can basically be divided into two categories: feature-level fusion [77,104,105] and decision-level fusion [106]. Feature-level fusion is mostly based on average pooling, weighted combination [107], serial combination, or concatenation of selected values. Liu et al. [77] utilized the latent semantic index method to represent multi-label information, while Yu et al. [49] constructed a novel parallel framework of attribute fusion to avoid the impact of duplicated information. This fusion level enhances the information from multiple sources and allows flexibility in fusion techniques, such as early integration, intermediate integration, and late integration [108]. But low data quality and difficulty in feature selection will affect building one efficient computational model. At the decision level, basic classifiers are used for different data sources, first for selecting the suitable ones; then, the results of each chosen method are ensembled as part of the determination protocol [109], as for the decision voting process [106]. Though the integration strategy is simple, this method can help create various decision-making systems that lead to more robust and accurate predictors. For instance, a multi-view model like ML-FGAT [76] incorporates most of the feature types (e.g., sequence, evolutionary information, physicochemical property, etc.), which minimizes the perturbation of extraneous data in predictive tasks while concurrently enhancing the descriptive capability. 

**Table 2 biomolecules-14-00409-t002:** A summary of state-of-the-art knowledge-based and fusion models for protein subcellular localization prediction. S: Single-Location; M: Multi-Location; Pub: Publication Cited; PsePSSM: Pseudo Position-Specific Scoring Matrix; PC: Physicochemical Properties; CT: Conjoint Triad; DE: Differential Evolution; wMLDAe: Weighted Linear Discriminant Analysis; F-GAN: Feature-Generative Adversarial Networks; GAT: Graph Attention Networks; KNN: K-Nearest Neighbor; CNN: Convolutional Neural Network; RF: Random Forest; CDD: Conserved Functional Domain; PseAAC: Pseudo Amino Acid Composition; PSSM: Position-Specific Scoring Matrix; NN: Nearest Neighbor; PPI: Protein-Protein Interaction Network; KEGG: KEGG (Kyoto Encyclopedia of Genes and Genomes) Pathway; mRMR: Minimum Redundancy Maximum Relevance; MCFS: Monte Carlo Feature Selection; LightGBM: Light Gradient Boosting Machine; IFS: Incremental Feature Selection; SVM: Support Vector Machine; SMOTE: Synthetic Minority Over-sampling Technique; EBGW: Encoding Based on Grouped Weight; RPT: Residue Probing Transformation; EDT: Evolutionary Distance Transformation; MCD: Multiscale Continuous and Discontinuous; MLSI: Multi-Label Information Latent Semantic Index; IRWLS: Newton-Weighted Least Squares Iterative Method; MLFE: Multi-Label Learning with Feature Induced Labeling Information Enrichment; DT: Decision Tree; DC: Dipeptide Composition; BR: Binary Relevance Method; CC: Classifier Chain; ECC: Ensemble Classifier Chain; SCF: Self-consistency Formulation; ML-KNN: Multi-Label K-Nearest Neighbor; FunD: Functional Domain; OET-KNN: Optimized Evidence-Theoretic K-Nearest Neighbor; SwissSCL: Swiss-Prot Subcellular Location Annotation; Acc: Accuracy; Prec: Precision; F1: F1 Score; HL: Hamming Loss; RL: Ranking Loss; OE: One Error; CV: Coverage; AT: Absolute Ture; AF: Absolute False; MCC: Matthews Correlation Coefficient; AUC: Area Under the Curve; OLA: Overall Location Accuracy; Rec: Recall.

Method	Features	Algorithm	S/M-Location	Species	Performance Metrics ^1^	Pub	Year
ML-FGAT	GO, PsePSSM, PC, CT,	DE, wMLDAe, F-GAN, GAT, KNN, CNN	M	Human, Virus, Gram-negative Bacteria, Plants, SARS-CoV-2	Acc: 0.91~0.96 Prec: 0.92~0.99 F1: 0.94~0.98 HL: 0.01~0.04 RL: 0.02~0.06 OE: 0.04~0.07	[76]	2024
PMPSL-GRAKEL	GO	RF, Random k-label sets algorithm	M	Human, Bacteria, Animal	Acc: 0.89~0.97 CV: 0.92~0.98 AT: 0.82~0.95 AF: 0.01~0.02	[89]	2024
Wang et al.	GO, CDD, PseAAC, PSSM	NN	M	Human	Acc: 0.84 F1: 0.76	[75]	2023
Zhang et al.	PPI, KEGG, GO	mRMR, MCFS, LightGBM, IFS, RF, SVM, SMOTE	M	Human	Acc: 0.75~1.00 MCC: 0.80~0.85	[105]	2022
ML-locMLFE	GO, PseAAC, EBGW, RPT, EDT, MCD	MLSI, IRWLS, MLFE	M	Bacteria, Plants, Virus	Acc: 0.94~0.99 Prec: 0.99~1.00 AUC: 0.98~0.99 OLA: 0.99~1.00 HL: 0.00~0.01 CV: 0.07~0.08 RL: 0.00	[77]	2021
Chen et al.	GO, KEGG, PPI, PC	RF, mRMR, IFS, SVM, KNN, DT, SMOTE	S	Human	Acc: 0.56~0.80 MCC: 0.49~0.76	[96]	2021
Gpos-ECC-mPLoc	GO, DC	BR, CC, ECC, SVM	M	Gram-positive Bacteria	Acc: 0.90~0.93	[110]	2015
mGOASVM	GO	SVM	M	Virus, Plants	Acc: 0.87~0.89	[92]	2012
iLoc-Euk	GO, PseAAC, PSSM, SCF	ML-KNN	M	Eukaryotes	Acc: 0.79	[111]	2011
Gneg-mPLoc ^2^	GO, FunD, PSSM	OET-KNN	M	Gram-negative Bacteria	Acc: 0.85~0.98	[85]	2010
PSORTb 3.0	SwissSCL	SVM	S	Eukaryotes, Prokaryotes	Acc: 0.97~0.98 Prec: 0.97~0.98 Rec: 0.93~0.94 MCC: 0.79~0.85	[112]	2010

^1^ The entries in this column are directly collected from the respective original publications. ^2^ Web server available at http://www.csbio.sjtu.edu.cn/bioinf/Gneg-multi/.

## 4. Bioimage-Based Methods

Imaging data show direct visual evidence of protein localization within different cell components, allowing precise and accurate location determination. Through imaging processing, computational models can analyze the spatial distribution of proteins at the single cell level and quantify their localization patterns. The complexity of images offers different levels of features, which also requires multiple preprocessing steps, deep classification models, and a longer running time to deal with for better performance.

### 4.1. Bioimage-Based Features

Compared to amino acid sequences, representing proteins with 2D images is more interpretable and concise when determining the subcellular localization. With the rapid improvement in microscopic imaging technology, scientists have paid more attention to bioimage-based methods. Computer hardware improvement, especially in graphics processing units (GPUs), makes it possible to deal with more complex calculation problems. The development of neural network structure also accelerates deep learning algorithm architecture improvement for image analysis significantly. For high-quality data, with the mission of mapping all human proteins in cells, tissues, and organs, the Human Protein Atlas (HPA) program [113] was initialized in 2003 as an open-access database that consists of imaging data, mass-spectrometry-based proteomics data, transcriptomics data, etc. The subcellular section of HPA shows detailed expressions and spatial distribution conditions of proteins encoded by 13,147 genes. As it recently updated to version 23, it is one of the most powerful training data sources for computational method development [19,114]. According to most recent studies, immunofluorescence (IF) images and immunohistochemistry (IHC) images are commonly selected as benchmark training and testing data sources.

The subcellular location features (SLF) collected can be divided into two categories, namely, global features and local features [115]. Composed of DNA distribution information and global textures, the global features such as morphological features, local binary patterns (LBP) [116] and Zernike features [117] mainly describe the spatial structure of the whole image. The Haralick [118] texture feature, which obtains statistical features including contrast, correlation, and entropy from the gray-level co-occurrence matrix of input images, is one well-known global image descriptor in pattern recognition. Local features, instead, can describe the micro-patterns ignored in global features. Take scale-invariant feature transform (SIFT) [119] as an example. SIFT was originally used for salient point detection and is suitable for fluorescence object description, which guarantees good performance in fluorescence image studies, especially when combined with global features. 

### 4.2. Bioimage-Based AI Methods

Image-related methods can be roughly organized into three phases based on the algorithms and the number of data types used, namely conventional or traditional machine learning methods, deep learning methods, and complex fusion methods, respectively. Figure 3 shows the development of these models from simple to complicated.

Traditional machine learning methods construct the prediction models with the aforementioned hand-crafted features for classification [120,121,122]. For instance, Li et al. [123] extended a logistic regression algorithm with structured latent variables for underlying components in different image regions for further classification. With two-layer deep-learned feature selection, Ulah et al. [124] established an SVM model based on both radial basis function and linear kernel for location prediction. However, these convolutional methods can be sensitive to noise and variability of imaging data collected, resulting in decreased model robustness. Spatial relationships embedded in images are rarely detected as well, due to manual feature engineering. As deep learning predictors are employed and have achieved high performance on various image-based tasks, recent advances in protein subcellular location rely more on deep learning methods [120]. 

Deep neural network implementation is the starting point, which increases the inner feature extraction power and the model’s learning ability for large and complicated datasets. In addition to selecting and integrating key features during the image preprocessing steps, most of the deep neural networks consider processed image segmentation as inputs for multi-layer convolutional neural networks (ML-CNN) [125]. Moreover, some predictors can integrate both low- and high-level features embedded in bioimages for a more in-depth view. For multi-label prediction, traditional CNN is extended with a criterion learning strategy to leverage label–attribute relevancy and label–label relevancy to determine the final location [126,127]. 

Implementing attention mechanisms is another successful attempt for image classification tasks [128]. With a conventional neural network backbone, Long et al. [129] introduced self-attention and multi-head attention layers as encoders to aggregate multiple feature vectors to construct a combined representation of all immunohistochemistry images input for subsequent analysis. Wang and Wei [126] applied Vision Transformer (ViT) [128] to learn multi-scale feature representations and integrate them globally before entering into the fully connected network. Through different types of transformers (e.g., vision, graph, resolution, etc.), Zhao et al. [130] optimizes the full extent of information embedded in the imaging data. However, there is still a lack of protein subcellular localization studies from this perspective, partly because it lacks efficiency compared to convolutional architectures at large-scale analysis.

In addition, the diversity in input data types across various dimensions shapes the complexity of the entire model. To be more specific, from image datasets, DeepPSL automatically learns meaningful features and their correlations for prediction improvement [131]. Xue et al. [132] unmixed the IHC images into protein and DNA channels for representation construction while segmenting the images into patches for fine-tuning network training. Ding et al. [133] ensemble different classification models using different depths of feature vectors constructed from images as inputs to achieve high-accuracy outputs. By collecting different imaging types, Wei et al. [134] built another parallel integrative deep network for label-free cell optical images. More details about the models can be found in Table 3. Though further techniques can be applied during the pretraining step [129,135,136,137,138], image-only methods still lack generalization capability and external validation. When incorporating greater modality of data that are not directly observable from imaging alone but related to protein subcellular localization during model establishment, it will take more contextual information into consideration and overcome the limitations in model performance. 

## 5. Protein Subcellular Localization in Different Species

Analyzing species separately allows a more accurate model generalization, since specific proteins and their subcellular localization patterns may differ in various cell organizations and organelle structures. Take bacteria as an example. As prokaryotes, they exhibit significant structural differences from eukaryotic organisms, like lacking common cellular organelles such as mitochondria, endoplasmic reticulum, and Golgi apparatus. However, within bacteria, a notable class of self-assembling microstructures, known as bacterial microcompartments (BMCs), consist of a protein shell encapsulating an enzymatic core [143,144], creating an internally enclosed space for protein to reside. Furthermore, bacteria possess special cell walls that can be classified as Gram-positive and Gram-negative bacteria [145], which are closely associated with different protein localization modes. For real-world application [146], the subcellular localization changes in host cells, like plants that need precise localization after viral infection, can give insights into the interactions of host cells and viruses, which helps in genetic resistance target identification [147]. 

Many models have been specially designed for distinct species (e.g., iLoc-Euk [111], iLoc-Virus [148], iLoc-Plant [149], and mPLR-Loc [150]). Gram-LocEN [151] is a predictor for large-scale datasets of both single- and multi-location proteins in bacteria. It created two databases called ProSeq and ProSeq-GO for query protein from Swiss-Prot and GOA databases [152], respectively, to guarantee the effectiveness and decrease storage complexity. After defining GO space and constructing GO vectors, the model demonstrated elastic net (EN) to enable automatic feature selection and further classification. DeepYeast [125], on the other hand, is a neural network trained specially for classifying fluorescent protein subcellular localization in yeast cells with images. As benchmark dataset construction is the foundation of building precise AI-based models, new methods tend to use datasets that have been collected and tested by previous models [28,92], like the Gram-positive and the Gram-negative bacteria dataset [153], the virus dataset [148,154], the plant dataset [149,155], the SARS-CoV-2 dataset [156], the animal cell lines [157], etc. Like Zou et al. [122], some models obtain the data directly by a manual literature search from UniProt and HPA database [70,112]. Multi-species database Compartments [158], fungal database FunSecKB2 [159], plant database PlantSecKB [160], and human and animal database MetazSecKB [161] mostly obtained and arranged from UniProt have also provided efficient searches for each organism and high-quality protein subcellular location annotation datasets across species. 

## 6. Current Challenges and Future Directions

### 6.1. Challenges

Despite the significant advances, challenges still exist for AI-based method development in the protein subcellular localization field. The interpretability of the model will be one of the big concerns. Actually, we have developed a series of interpretable machine learning approaches [162,163,164,165] for protein subcellular localization and membrane protein function prediction. However, most of them are based on linear models. As deep learning algorithms have complicated training processes that generate high dimensional and nonlinear deep features for prediction, it is of great importance to interpret the decision-making procedures of the model for a better understanding of the essential factors that influence protein localization. SHAP [166], DeepExplainer [167] based on DeepLift [168], and other methodologies major in capturing the importance of features for overall prediction tasks have been implemented in recent studies for increasing model interpretability. Luo et al. [69] have also reduced the dimensionality of feature vectors by constructing autoencoders to obtain a better feature representation for downstream analysis. In ML-FGAT [76], the interpretability is strengthened by analyzing the attention weight parameters. Explainable and understandable frameworks will give more reliable predictions that benefit further studies from a biological perspective.

Moreover, protein subcellular location is influenced by multiple factors. AI-based methods mostly rely on original sequences or images as inputs, which lack the information after protein biosynthesis. There is also a chance that the prediction model provides the same subcellular location when the mutant protein resides in a different place [147]. Post-translational modifications (PTMs), which refer to amino acid side-chain modification after the synthesis of some proteins, can contribute significant changes to their subcellular location [169]. There are many kinds of PTMs, such as phosphorylation, glycosylation, and acetylation, which dynamically regulate the protein within the cell simultaneously [170], resulting in sparse and incomplete experimental data for model training. As more post-translational positions are discovered [170], AI-based predictions that consider PTMs as key features can also be further investigated [170,171].

Establishing models to leverage both annotated and unannotated proteins for localization can also be a challenge, with a large proportion of unreviewed data reported each year (Figure 1A,B). Though data augmentation methods like SMOTE and GAN are widely used to handle data imbalance, semi-supervised learning can also be established to solve the problem [138,172]. To be more specific, EnTrans-Chlo [173] incorporates multi-modal features and converts them into sample-to-sample similarity features with assigned weights for feeding a highly efficient learning model. LNP-Chlo [174] extended the previous approach by adopting a quadratic programming algorithm to optimize the weights of nearest neighbors. These semi-supervised models remarkably outperformed state-of-the-art supervised methods when integrating different data modalities and dimensionalities with less of a requirement for sufficient labeled data.

### 6.2. Future Directions

Currently, cutting-edge research directions in subcellular localization mainly lie in spatial proteomics [9] and RNA subcellular localization. 

With the blooming of single-cell research, it is possible to gain a full understanding of disease from cell and tissue heterogeneity. Since the exact location of proteins at the subcellular, cellular, or tissue levels directly links to their functions, it is essential for protein localization with a single-cell and spatial resolution [18]. Zhu et al. [175] have created cell-based methods with a pseudo-label assignment to discover protein subcellular localization results across distinct cells with heterogeneity among single cells. Husain et al. [140] presents the Hybrid subCellular Protein Localiser (HCPL) that robustly localizes single-cell subcellular protein patterns. Wang’s work with mass spectrometry (MS)-based spatial proteomics [176] shows the possibility of larger dimensional feature maps and higher learning ability of computational models. 

System-wide studies of RNA subcellular localization (e.g., mRNA [177]) have also paved the way for a more comprehensive analysis of the cellular dynamics [178,179], as proteins are usually transcribed by RNA molecules. Moreover, except for RNA transcripts for protein, other RNAs, like long noncoding RNAs (lncRNAs), may also be involved in many biological functions [180]. Predicting their subcellular locations with AI-based methods [180] can significantly reduce costs and time expenditure, enabling the investigation of their functionalities with limited data [178]. In addition, common [181] and rare cellular-compartment-specific prediction models can be further explored [182]. As for the data imbalance issue, most of the prediction models mainly focus on some of the subcellular components, since they have more manually adjusted records for model construction. 

In addition, other promising future directions in this field include web server or tool/software development for protein subcellular localization. Though accurate and efficient models are continuously published, only a few are freely available to the public. Moreover, since models are getting complex with multiple processing blocks, developing methods and/or algorithms into a web-based platform [53] or software service [183] would greatly facilitate experimental research and interdisciplinary collaboration. For downstream applications, Wang et al. [100] have detected mislocated proteins under drug treatments with established models. Xue et al. [132] developed a machine-learning model and validated its ability by identifying biomarker proteins related to colon cancer. Pang et al. [52] proposed the CNN-XGBoost model for Alzheimer’s Disease and achieved competitive performance among general methods. As model accuracy and consistency have been greatly increased, it will be more beneficial to apply complete models for different biomedical scenarios. 

## 7. Conclusions

In this review, we have reviewed three types of computational methods using machine learning or deep learning models to construct predictors for protein subcellular localization. For different kinds of inputs, such as protein sequence, GO terms, or IHC images, the predictors will first convey the biological data to numerical or mathematical representations of essential features embedded in the source and apply widely used classifiers for single or multi-class tasks. Traditional machine learning methods can combine various features and manage the high-dimensional data by dimensionality reduction techniques like random projection [184] to avoid the curse of dimensionality and achieve interpretable outcomes under large data scales. Alternatively, they can combine the results of different classifiers, which run the calculation parallelly, to improve the overall performance. Deep learning methods that are mostly based on neural networks will learn and extract high-level features and their correlations from the inputs before the classification. When dealing with large-scale datasets, prediction with a language model is also available with deep learning. For future direction, in addition to faster and more effective algorithm development, we also assume that the localization prediction will incorporate more biochemical interactions like protein–protein interaction networks (PPI), metabolic networks, gene co-expression interaction, etc., into consideration, since proteins intricately engage in complex physiological reactions within the cellular space. Above all, we are confident that the computational methods will raise more and more attention for (1) systematic research like proteomics and metabolomics, (2) provide dynamic insights into cells and reveal what the influence will be when the target protein is muted; and (3) assist the experimental side with data analysis, experimental design, and so on. In the long run, this research area will benefit clinical drug development and contribute to disease detection, diagnosis, prognosis, and treatment.

## Figures and Tables

**Figure 1 biomolecules-14-00409-f001:**
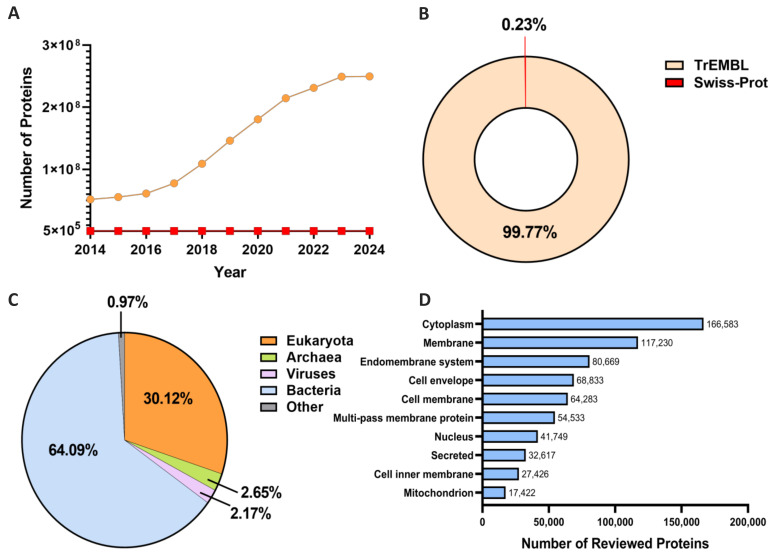
Statistical analysis of UniProtKB [17] (2024_01.version). (**A**) The trend of protein number growth in TrEMBL (unreviewed proteins) and Swiss-Prot (reviewed proteins). The number of newly discovered unannotated proteins far exceeds that of newly added experimentally validated proteins. (**B**) The proportion of newly added protein counts between the two databases in the 2024_01.version. (**C**) Taxonomic distribution of protein sequences. (**D**) Number of proteins in the top 10 subcellular locations.

**Figure 2 biomolecules-14-00409-f002:**
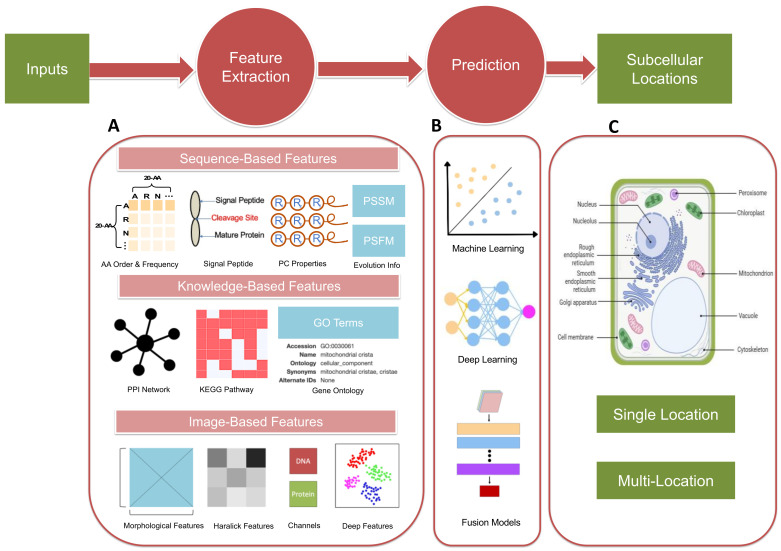
The flowchart of three major types of AI-based prediction methods. The procedures include sequences or images as input, feature extraction, model prediction, and subcellular location output. (**A**) Key features extracted from sequences, annotations, and image inputs. Different classifiers extract composition information, encompassing AA order and frequency, physicochemical properties, and identifying signal peptide cleavage sites from sequence inputs. In addition to straightforward data, evolutionary profiles are also considered through homology alignment with the Position-Specific Scoring Matrix (PSSM) and the Position-Specific Frequency Matrix (PSFM). Knowledge-based methods involve the establishment of Gene Ontology (GO) vectors, derived from GO terms collected from specific databases with protein sequences or accession numbers as keywords. Other functional annotations, such as protein–protein interaction (PPI) and Kyoto Encyclopedia of Genes and Genomes (KEGG) pathway information, can also be fused as input features. Imaging features mainly consist of morphological, Haralick data and information from different channels, namely hand-crafted features, and deep features captured by deep learning algorithms. (**B**) Three types of algorithms used for prediction modules in computational models. (**C**) Major subcellular locations in a plant cell as an example of potential outputs for proteins with single or multiple locations.

**Figure 3 biomolecules-14-00409-f003:**
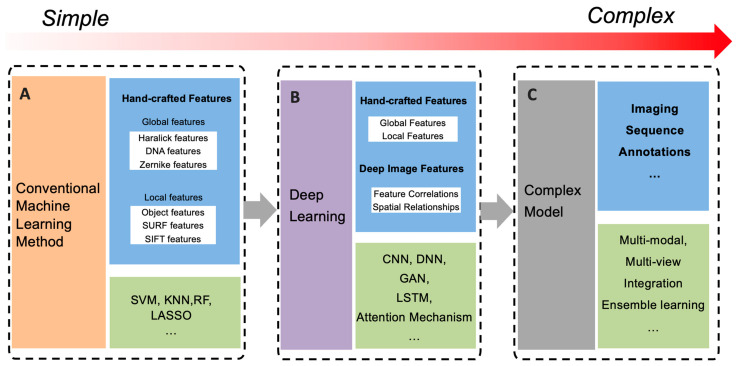
Three primary categories of computational methodologies for processing imaging data. The red arrow depicts the progressive complexity of prediction models, reflecting advancements toward more sophisticated computational frameworks. Blue rectangle: features used for model training; green rectangle: algorithms for location prediction. (**A**) Conventional Machine Learning Methods. Hand-crafted figures representing global and local information of images are extracted and trained for simple models. (**B**) Deep Learning Methods. Coupled with hand-crafted features, deep image features are obtained by deep neuro networks. (**C**) Complex Fusion Models. This method integrates multi-modality data like sequence, annotation texts, and imaging data as model inputs to gain a more comprehensive and interpretable model for protein subcellular localization. SURF: Speeded Up Robust Features. SIFT: Scale-Invariant Feature Transform. SVM: Support Vector Machine. KNN: K-Nearest Neighbor. RF: Random Forest. LASSO: Least Absolute Shrinkage and Selection Operator. CNN: Convolutional Neural Network. DNN: Deep Neural Network. GAN: Generative Adversarial Network. LSTM: Long Short-Term Memory.

**Table 1 biomolecules-14-00409-t001:** A summary of state-of-the-art sequence-based protein subcellular localization frameworks. S: Single-Location; M: Multi-Location; Pub: Publication Cited; BERT: Bidirectional Encoder Representations from Transformers; XGBoost: Extreme Gradient Boosting; GAN: Generative Adversarial Network; CNN: Convolutional Neural Network; LM: Language Model; MLP: Multilayer Perceptron; SP: Signal Peptide; PC: Physicochemical Properties; PSSM: Protein-Specific Scoring Matrix; LSTM: Long Short-Term Memory; CTM: Consensus Sequence; AECA: Absolute Entropy Correlation Analysis; LDA: Linear Discriminant Analysis; SVM: Support Vector Machine; MAM: Multi-Attention Mechanism; PseAAC: Pseudo Amino Acid Composition; SAAC: Split Amino Acid Composition; KNN: K-Nearest Neighbor; AAF: Amino Acid Frequencies; GCF: Gene Co-expression Features; DNN: Deep Neural Network; AAC: Amino Acid Composition; Acc: Accuracy; Prec: Precision; Rec: Recall; F1: F1 Score; GM: Grand Mean; MicroF1: MicroF1 Score; MacroF1: MacroF1 Score; MCC: Matthews Correlation Coefficient; Jaccard: Jaccard Value; AUC: Area Under the Curve; Spec: Specificity; Sen: Sensitivity; FPR: The False Positive Rate; HL: Hamming Loss; RL: Ranking Loss; OE: One Error; CV: Coverage.

Method	Features	Algorithm	S/M-Location	Species	Performance Metrics ^1^	Pub	Year
DaDL-SChlo	Deep- and Hand-crafted features	ProtBERT, XGBoost, GAN, CNN	M	Plants	Acc: 0.86~0.94 Prec: 0.88~0.95 Rec: 0.86~0.94 F1: 0.86~0.95 GM: 0.84~0.94	[61]	2023
DeepLoc—2.0	Masked-LM Objective	MLP, Protein LM	M	Eukaryotes	Acc: 0.39~0.73 MicroF1: 0.60~0.73 MacroF1: 0.46~0.66 MCC: 0.17~0.90 Jaccard: 0.53~0.69	[70]	2022
SignalP—6.0	SP	Transformer Protein LM	M	Archaea, Gram-positive Bacteria, Gram-negative Bacteria and Eukaryotes	MCC: 0.65~0.89 Prec: 0.53~0.94 Rec: 0.50~0.88	[28]	2022
MULocDeep ^2^	PC, PSSM	LSTM	M	Viridiplantae, Metazoa, Fungi	AUC: 0.74~0.95	[71]	2021
SCLpred-EMS ^3^	Sequence Motifs	Deep N-to-1 CNN	S	Eukaryotes	MCC: 0.75~0.86 Spec: 0.89~0.97 Sen: 0.75~0.89 FPR: 0.02~0.05	[53]	2020
CTM-AECA-PSSM-LDA	CTM, AECA-PSSM	LDA, SVM	S	Apoptosis Proteins on CL317 and ZW225 datasets	Acc: 0.95~0.99 MCC: 0.90~1.00 Spec: 0.94~1.00 Sen: 0.91~0.95	[36]	2020
TargetP—2.0	SP	LSTM, MAM	S	Plants and Non-plants	Prec: 0.75~0.98 Rec: 0.75~0.98 F1: 0.75~0.98 MCC: 0.75~0.97	[27]	2019
Javed and Hayat	PseAAC, SAAC	ML-KNN, Rank-SVM	M	Bacteria, Virus	Acc: 0.80~0.85 Prec: 0.88~0.90 HL: 0.07~0.09 RL: 0.07~0.08 OE: 0.17~0.20 CV: 0.26~0.51	[35]	2019
MU-LOC ^4^	AAF, PSSM, GCF	DNN, SVM	S	Plants (Mitochondrian)	Acc: 0.74~0.94 Prec: 0.74~0.82 MCC: 0.50~0.67 Spec: 0.88~0.97 Sen: 0.60~0.70	[72]	2018
MultiP-SChlo	PseAAC	SVM	M	Plants (Subchloroplast)	Acc: 0.55~0.60 Prec: 0.64~0.65 Rec: 0.66~0.71 F1: 0.65~0.67	[73]	2015
SlocX	AAC, Gene Expression Profile	SVM	S	Plants	Prec: 0.83 MCC: 0.48 Sen: 0.33	[74]	2011

^1^ The entries in this column are directly collected from the respective original publications. ^2^ Web server available at http://mu-loc.org. ^3^ Web server available at http://distilldeep.ucd.ie/SCLpred2/. ^4^ Available at http://mu-loc.org.

**Table 3 biomolecules-14-00409-t003:** A summary of state-of-the-art image-based methods for protein subcellular localization prediction. S: Single-Location; M: Multi-Location; Pub: Publication Cited; LBP: Local Binary Pattern; PSSM: Position-Specific Scoring Matrix; PseACC: Pseudo Amino Acid Composition; PC: Physicochemical Properties; LASSO: Least Absolute Shrinkage and Selection Operator; BR: Binary Relevance; SDA: Stepwise Discriminant Analysis; CNN: Convolutional Neural Network; MSA: Multihead Self-attention; Swin: Swin Transformer; CAFE: Cross Attention Feature Enhancement; DNN: Deep Neural Network; CLH: Cell-level Hybrid Model; CLA: Cell-level Actnet; VID: Visual Integrity Detector; ResNet: Residual Network; SE: Squeezeand-Excitation; DenseNet: Dense Convolutional Network; MIL: Multi-instance Learning; SRS: Stimulated Raman Scattering; MPFNet: Multiple parallel Fusion Network; MLP: Multi-Layer Perceptron; SLFs: Subcellular Location Features; CLBP: Completed Local Binary Pattern; LETRIST: Locally Encoded Transform Feature Histogram; RICLBP: Rotation Invariant Co-occurrence Among Adjacent Local Binary Patterns; GDA: Generalized Discriminant Analysis; DCF: Deep-cascade Forest; IF: Immunofluorescence Microscopic; GNT-Xent: The Gradient-Stabilized and Normalized Temperature-Scaled Cross-Entropy Loss; Acc: Accuracy; Prec: Precision; Rec: Recall; MAE: Mean Absolute Error; NRMSE: Normalized Root Mean Square Error; SSIM: Structural Similarity Index; PCC: Pearson’s Correlation Coefficient; R2: Coefficient Determination; F1: F1 Score; MicroF1: MicroF1 Score; MacroF1: MacroF1 Score; Dice: Dice Similarity Coefficient; mIOU: The Mean Intersection Over Union (IOU); MCC: Matthews Correlation Coefficient.

Method	Features	Algorithm	S/M-Location	Species	Performance Metrics ^1^	Pub	Year
Zou et al.	Haralick, LBP, PSSM, PseAAC, PC	LASSO, BR, SDA, CNN	S	Human	Acc: 0.75~0.86 Prec: 0.80~0.85 Rec: 0.74~0.85	[122]	2023
ST-Net	Low- and High-Level features	MSA, Swin, CAFE, CNN,	S	Human	MAE: 0.15~0.23 NRMSE: 0.30~0.31 SSIM: 0.78~0.89 PCC: 0.94~0.95 R2: 0.87~0.88	[139]	2023
HCPL	Cell- and Image-Level Information	DNN, CLH, CLA, VID	M	Human	Prec: 0.55~0.57	[140]	2023
Ding et al.	Features Generated from ResNet	ResNet-34, SE, GAP-net, DNN	M	Yeast	Acc: 0.91 Prec: 0.89 Rec: 0.90 F1: 0.89	[133]	2023
Muti-task Learning Strategy	Features Generated from ResNet and DenseNet	ResNet, DenseNet, MIL, CNN	M	Human	MicroF1: 0.78 MacroF1: 0.71	[135]	2022
MPFnetwork	SRS and Fluorescence Signal	MPFNet, CNN, MSA, MLP	M	Human	NRMSE: 0.19~0.20 SSIM: 0.89~0.92 PCC: 0.90~0.91 Dice: 0.93~0.94 mIOU: 0.87~0.88	[134]	2022
PScL-DDCFPred	SLFs, LBP, CLBP, LETRIST, RICLBP	SDA-GDA, DNN-DCF	M	Human	Acc: 0.88 Rec: 0.88 Prec: 0.89 F1: 0.88 MCC: 0.86	[141]	2022
PLCNN	Image block structure	CNN	M	Human, Yeast	Acc: 0.91~1.00	[142]	2022
SIFLoc	IF images	GNT-Xent, RandAugment, ResNet18	M	Human	Acc: 0.67~0.73 Prec: 0.77~0.81 Rec: 0.69~0.74 F1: 0.73~0.77	[137]	2022
DeepYeast	Haralick, Gabor, Zernike Features	CNN, DNN	M	Yeast	Acc: 0.97~0.99 Prec: 0.70~0.95 Rec: 0.65~0.92	[125]	2017

^1^ The entries in this column are directly collected from the respective original publications.

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
