# Peer review of "A Review for Artificial Intelligence Based Protein Subcellular Localization"

_biomolecules, 2024, doi:10.3390/biom14040409_

Round 1

Reviewer 1 Report

Comments and Suggestions for Authors

This review of protein subcellular localization prediction methods includes a lot of references to published methods.  However, there is NO comparison of performance of methods.  But worse, there are gross errors in the main Table of results, making it hard to trust any of the review as comprehensive and accurate.

Very Major:

1) There is no comparison of performance of the different published methods.  Tables 1, 2,and 3 should at least include some metrics of prediction performance.  This may not be uniform, since the different publications likely provided different performance metrics; however, most should include standard metrics like accuracy, precision, recall, F1-score, MCC, etc.  But without some review of performance, this review is simply a bucket of methods with no insight into how methods and performance have improved and which methods provide superior performance.

2) Some of the attributions of machine learning methods for specific published methods are just WRONG or grossly incomplete.  For example, Zhang et al, 2022 listed in Table 2 using Random Forest, SVM, and gradient boosted trees.  Also, the paper is focused on subcellular-informing feature selection for future methods development. All methods described in Tables 1, 2, and 3 MUST be reviewed again and the table entries corrected. 

Zhang, Yu-Hang, et al. "Subcellular localization prediction of human proteins using multifeature selection methods." BioMed Research International 2022 (2022).

Another example in Table 2 is Chen et al, 2023 with reference “[102]”.  The entry in Table 2 lists the publication date as 2024.  The “[102]” reference has the following 2023 date:

Chen, L.; Qu, R.; Liu, X. Improved Multi-Label Classifiers for Predicting Protein Subcellular Localization. MBE 808 2023, 21, 214–236, doi:10.3934/mbe.2024010.

However, Google Scholar gives the following citation year of 2022:

Chen, Lei, Ruyun Qu, and Xintong Liu. "Improved multi-label classifiers for predicting protein subcellular localization." Mathematical Biosciences and Engineering: MBE 21.1 (2022): 214-236.

Here is another example in Table 3, Zou et al. 2023, which uses a convolutional neural network, but the entry indicates LASSO.

Zou, Kai, et al. "Dual-Signal Feature Spaces Map Protein Subcellular Locations Based on Immunohistochemistry Image and Protein Sequence." Sensors 23.22 (2023): 9014.

Major:

1) Introduction mentions UniProt, but not the Human Protein Atlas, which has the most comprehensive subcellular localization information, at least for human cells.  Part of this is likely due to describing the problem with respect to the whole biosphere.  

2) It should be better explained why Swiss-Prot is much smaller than TrEMBL, which is due to the manual curation step required.  But one could argue that manual curation is not required, especially if the transcript translated sequences are cross-validated with proteomics detection.  HPA did the reverse in using RNAseq to corroborate their immunofluorescence subcellular localization data.

3) There are no references to the BioPlex interactome papers, which provide extensive interactome information for the knowledge-based features and methods.

4) There is NO discussion of gold-standard datasets needed for methods development.

Minor:

Line 311: “network” should be plural: “networks”.

The Tables include acronyms that are not defined in the table legend or footnote.

Comments on the Quality of English Language

English is mostly fine.

Reviewer 2 Report

Comments and Suggestions for Authors

The paper reviews the latest advances in artificial intelligence (AI) and machine learning (ML) methods for predicting protein subcellular localization. It discusses three major approaches: sequence-based, knowledge-based, and image-based methods. Sequence-based methods analyze amino acid sequences, evolutionary information, and sorting signals. Knowledge-based methods utilize Gene Ontology (GO) terms, protein-protein interactions, and pathway information. Image-based methods extract features from microscopy images. The paper also covers challenges such as improving interpretability, considering post-translational modifications, and handling unannotated proteins. Future directions include spatial proteomics and RNA subcellular localization. The authors emphasize the potential of computational methods to assist in systematic research, provide dynamic insights into cells, and benefit drug development and disease diagnosis.

However, I have the following concerns,

(1) I notice that only three references are provided regarding Transformer-based deep learning methods. Given the remarkable performance demonstrated by AlphaFold, it would be beneficial for the author to delve deeper into the usage, advantages, and limitations of Transformer methods. Additionally, considering that Transformer-based methods are increasingly becoming the preferred choice in the computer vision community, it is essential to include a discussion of their relevance and potential applications within the image-based method section.

(2) The discussion provided by the authors regarding future directions appears to be rather brief, which may not be optimal for a review article. A key aspect of review articles is the exploration of future directions in the field. Readers typically expect insights and discussions on potential areas of importance, emerging trends, and avenues for further research. Thus, it would be beneficial for the authors to expand on their discussion of future directions to provide a more comprehensive outlook on the field's trajectory.

(3) Please consider indenting the captions of all figures and tables, or using italicized text for captions. Currently, they are at the same indentation level as the main text, which can make it difficult to distinguish them, especially from line 99 to 112. This can cause confusion, as readers may mistake them for part of the paper's text when they are actually captions for Figure 2.

Comments on the Quality of English Language

English looks good to me.

Reviewer 3 Report

Comments and Suggestions for Authors

Wan and his colleagues examine the most recent developments in AI-based technique creation in this article, covering three common categories of approaches: sequence-based, knowledge-based, and image-based methods. The authors also go into great detail about the difficulties that currently exist and potential paths for developing AI-based methods in this area of study. Following the introduction to the topic, the author presents the methods developed based sequence, knowledge and imaging techniques. The list methods presented in each section is interesting and a valuable addition to the literature. The figures in the paper provide overall understanding of AI based methods development and challenges.

In general, the review is well organized and presented in detail about the state of the art techniques. This review will definitely impact the field of AI in healthcare. I recommend the paper for publication after addressing couple of my minor comments.

However, the authors made a comprehensive review on the topic presented, I think the authors should more emphasize and illustrate their own idea on AI based protein subcellular localization.

Review is lack of detailed summary of the advantages and disadvantages of various AI based models.

Round 2

Reviewer 1 Report

Comments and Suggestions for Authors

This reviewer is pleasantly surprised by the huge improvement made to the main results in Tables 1, 2, and 3. The Algorithms mentioned in these tables are much more comprehensive and the addition of Features and Performance Metrics now provides real value to the review.

The other additions to the main text provide a better description of relevant data sources and methods as well.

The authors have addressed my concerns.